# Factors Affecting the Infection Control Practices of Nurses at University Hospitals

**DOI:** 10.3390/healthcare10081517

**Published:** 2022-08-12

**Authors:** Mi Hyang Lee, Sun Hwa Jun

**Affiliations:** Department of Nursing, Konyang University, Daejeon 35365, Korea

**Keywords:** nursing professionalism, time pressure, infection control, organizational culture, practice

## Abstract

This study was conducted to check the extent of nursing professionalism, time pressure, infection control, organizational culture, and the infection control practices of nurses, and to assess the factors that impart an influence on their infection control practices. This is a descriptive survey study aimed at the assessment of factors that impart an influence on the infection control practice of nurses by using a structuralized questionnaire. As the result of this study, the infection control practices of nurses have a positive correlation with the time pressure (r = 0.16, *p* = 0.034) and the organizational culture for infection control (r = 0.29, *p* < 0.001). Factors that affect the infection control practices included the organizational culture for infection control (β = 0.29, *p* < 0.001) and time pressure (β = 0.16, *p* = 0.024), with the explanation power of 10%. It was possible to confirm that the affirmative organizational culture for infection control plays an important role in enhancing the infection control practices of nurses. Accordingly, it is necessary to provide administrative and financial support from the organization, including support by the management and administrators of clinical practices, as well as the provision of required commodities in order for nurses to execute infection control in accordance with the prescribed regulations.

## 1. Introduction

Medical-related infection increases the economic burden on patients due to the increase in the treatment costs and extension of the hospitalization period of patients [1]. It is therefore very important for healthcare providers to conduct their respective duties in accordance with infection management guidelines to minimize the manifestation of infection within the medical institution and, in particular, nurses who have the closest contact with patients among healthcare providers have the responsibility to provide high-quality nursing services to the patients under an environment that is safe from infection [2]. However, it has been found that an excessive workload and a lack of time, etc., which nurses experience [3,4], impart an adverse effect on their infection management practices.

Factors that affect the level of infection management practice include personal characteristics, such as nursing professionalism, moral sensitivity and level of awareness, etc. [5,6], and organizational characteristics, such as an organizational culture within the medical institution [7]. Among these, nursing professionalism is an important factor to improve the level of nursing professionals and for nurses to provide high-quality nursing services as a profession [2]. Desirable nursing professionalism is a concept through which nurses are not only able to experience personal fulfillment, but also that the value of nursing is acknowledged by the public [8]. Nursing professionalism imparted affirmative influence on COVID-19 nursing practices [9].

Given the characteristics of nursing tasks, nurses experience time pressure quite frequently due to the burden of having to complete a wide range of tasks related to the patient within a limited time, while conducting indirect nursing tasks, such as goods management and keeping medical records, etc., in addition to direct nursing tasks, such as the administration of medicine during their work hours [3]. The increase in time pressure for the requirement of having to complete nursing tasks within a prescribed time imparts a greater adverse effect on safe nursing activities for patients [3,4]. The excessive workload increases the burden on nurses in having to complete nursing tasks within a prescribed time, which, in turn, increases the job stress of nurses and a decline in the quality of nursing services provided to the patients [10]. Therefore, it is necessary to assess the effects of time pressure perceived by nurses under the current situation in which the infection management prevention guidelines have been fortified due to the COVID-19 pandemic on the level of infection management practices.

The support of superiors for infection management activities in and an organizational infection management culture related to infection management operations within the medical institution imparts an affirmative influence on the level of infection management practice of nurses [11]. The attitude of administrators towards infection management within the organization directly affects the infection management activities of nurses [12]. As such, it is important for the administrators of medical institutions to have volition and provide support for a safe environment, and for peers and superiors to provide affirmative feedback on the safety management performances of nurses [13]. It is necessary for the administrators to establish guidelines for nurses in implementing infection management practices and to establish an organizational infection management culture to ensure the provision of necessary resources for infection prevention and continuous infection management education as well as training, in order to prevent medically related infections [11].

After the COVID-19 pandemic, the importance of infection management within medical institutions has been fortified. Moreover, it is anticipated that there will be extensive changes in the medical institution environment related to infection management, thereby needing an assessment of the level of infection management practice of nurses and factors that affect such practices.

Accordingly, this study is aimed at the provision of basic data for the development of the infection control program for the improvement of the level of infection control practice of nurses by assessing the factors that affect the level of infection control practice of nurses working at university hospitals under the circumstances of the COVID-19 pandemic.

This study is aimed at the assessment of nursing professionalism, time pressure, organizational infection management culture and level of infection management practice of nurses, and assessment of factors that affect the level of infection management practice of nurses. Specific goals are as follows:(1)Assess nursing professionalism, time pressure, organizational infection management culture and level of infection management practice of nurses.(2)Assess the differences in the level of infection management practice in accordance with the general characteristics of nurses.(3)Assess the correlation between nursing professionalism, time pressure, and the organizational infection management culture level of infection management practice of nurses.(4)Assess factors that affect the level of infection management practice of nurses.

## 2. Materials and Methods

### 2.1. Study Design

This is a descriptive investigative study aimed at the assessment of nursing professionalism, time pressure, organizational infection management culture and level of infection management practice of nurses working at a university hospital, and assessment of factors that affect their level of infection management practice.

### 2.2. Subjects

The subject institution of this study is a university hospital with 800 hospital beds that operates a negative-pressure isolation room for air-mediated infection patients, a ward used exclusively for COVID-19, and an isolation room for contact or droplet infection patients. It has made substantial contributions toward blocking the proliferation of infection in the local society from the time of the outbreak of the new influenza virus in 2009 to the present, and it is in charge of practical education for a wide range of health professions as a university hospital.

Nurses working in departments in which infectious disease patients, including multi-drug resistant bacteria, air-mediated diseases and new infectious diseases, etc., are hospitalized, were selected arbitrarily through discussion with the head of the nursing department at the corresponding university hospital.

The number of subjects was computed by using the G*power 3.1.9.7 program to determine the sample size of the subjects. The number of study subjects is 157 when the median-effect size, significance level, test power and predictor variables are set at 0.15, 0.05, 0.90 and 12, respectively, based on the standards of the regression analysis method in accordance with the preceding study [14], and the questionnaire survey was distributed to a total of 189 subjects with the consideration for the dropout rate of 20%. Among these, 186 questionnaires were retrieved for a retrieval rate of 98%, and 183 questionnaires, after having excluded 3 questionnaires with insufficient answers, were used for final analysis.

### 2.3. Measures

#### 2.3.1. General Characteristics

The general characteristics of the subjects of this study were measured with a total of 9 questions on gender, age, education, marital status, religion, work experience, position, affiliate department and infection control education experience by referring to the preceding study.

#### 2.3.2. Nursing Professionalism

Nursing professionalism was measured by using the instrument that was developed by Yeun, Kwon and Ahn [14] and revised and supplemented by Han and Kim [15]. The content validity of the tool used in this study was verified by 3 professors at the nursing department and 2 heads of the nursing department, and the Content Validity Index (CVI) of more than 0.80 was measured for each question. The questionnaire was composed of a total of 18 questions in 5 subordinate domains, including 6, 5, 3, 2 and 2 questions on the self-concept of the profession, social awareness, nursing professionalism, practical nursing role and independence of nursing, respectively. The subjects were required to allocate scores using the 5-point Likert scale ranging from 1 for ‘Not at all’ to 5 for ‘Very much so’. A higher score signifies higher nursing professionalism. In a study by Yeun et al. [14], which developed the tool, the construct validity and Cronbach’s alpha as the reliability were 0.93 and 0.92, respectively, and, in the study by Han and Kim [15], measured by means of Cronbach’s alpha, it was 0.94, and Cronbach’s alpha of this study was 0.86.

#### 2.3.3. Time Pressure

Time pressure was measured by using the instrument developed by Putrevu and Ratchford [16], corrected as well as supplemented by Teng, Shyu, Chiou, Fan and Lam [17] for the use on nurses and translated by Yun and Son [3]. The content validity of the tool used in this study was verified by 3 professors at the nursing department and 2 heads of the nursing department, and the Content Validity Index (CVI) of more than 0.80 was measured for each question. The questionnaire is composed of a total of 5 questions, and subjects were required to allocate scores for each question using the 7-point Likert scale ranging from 1 for ‘Always not so’ and 7 for ‘Always so’. A higher score signifies a higher time pressure. The reliability in the study by Yun and Son [3] measured by means of Cronbach’s alpha was 0.93, and Cronbach’s alpha of this study was 0.93.

#### 2.3.4. Organizational Infection Management Culture

The organizational infection management culture was measured by using the instrument corrected and supplemented by Moon and Jang [11] based on the patient safety culture measurement instrument of Park [12]. The content validity of the tool used in this study was verified by 3 professors at the nursing department and 2 heads of the nursing department, and the Content Validity Index (CVI) of more than 0.80 was measured for each question. The questionnaire is composed of a total of 10 questions, and subjects were required to allocate scores for each question using the 7-point Likert scale ranging from 1 for ‘Not at all’ to 7 for ‘Very much so’. A higher score signifies a more affirmative organizational infection management culture for infection management practices. The reliability in the study by Moon and Jang [11] measured by means of Cronbach’s alpha was 0.86, and Cronbach’s alpha of this study was 0.85.

#### 2.3.5. Nursing Practice of Infection Control

The level of infection control practices was measured by using the instrument developed by Kim [18] and corrected and supplemented by Hong and Park [19]. The content validity of the tool used in this study was verified by 3 professors in the nursing department and 2 nurses specializing in infection control, and the Content Validity Index (CVI) of more than 0.80 was measured for each question. The questionnaire is composed of a total of 37 questions in 6 subordinate domains, including 9, 9, 7, 5, 4 and 3 questions on hand hygiene, infection control of catheters in blood vessels, urinary tract infection control, pneumonia control, isolation, and disinfection and sterilization control, respectively. The subjects were required to allocate scores for each question using the 5-point Likert scale ranging from 1 for ‘Not at all’ to 5 for ‘Always so’. A higher score signifies a higher level of infection control practices. Reliability in the study by Hong and Park [19] measured by means of Cronbach’s alpha was 0.85, and Cronbach’s alpha of this study was 0.97.

### 2.4. Data Collection

This study was deliberated and approved by the institutional review board (IRB) (*** 2021-06-020-001). Data were collected from 22 November to 10 December 2021. After having attained the approval of the nursing department of a university hospital in the D region for data collection and providing an explanation of the purposes and methods of the study to the head of the nursing department by the investigator, the questionnaire survey was conducted on the subjects who submitted a written consent for the participation in the study. It was explained to the study subjects that they can withdraw their consent to participate in the study at any time and that the results of the study will not be used for any purposes other than those stipulated for the study. Questionnaires were retrieved by providing an individual with an envelope to ensure the anonymity of the subjects. It takes approximately 15~20 min to complete the questionnaire, and a small gift was given out to subjects in appreciation of their participation.

### 2.5. Data Analysis

The statistical analysis was conducted on the data collected by using the SPSS/WIN 21.0 statistics program. First, the general characteristics, nursing professionalism, time pressure, organizational infection management culture and level of infection management practice of nurses were analyzed by using descriptive statistics. Second, differences in the level of infection management practice in accordance with the general characteristics of nurses were analyzed by using an independent *t*-test and one-way ANOVA, along with the use of the Scheffe test for the post hoc comparison. Third, the correlation among the study variables was analyzed by means of Pearson’s correlation coefficient. Fourth, factors that affect the level of infection management practice of nurses were analyzed by using multiple regression analysis.

## 3. Results

### 3.1. General Characteristics of Nurses and Differences in Their Performances in Accordance with Such General Characteristics

In terms of the gender of the nurses, females accounted for the larger proportion at 171 (93.4%), and the number was largest at 111 in the age bracket of 20~29 years (60.7%). In terms of academic background, 153 nurses (83.6%) had a bachelor’s degree and, in terms of marital status, 131 nurses (71.6%) were single. A total of 133 (72.7%) had no religion, 126 (68.9%) had more than 3 years of work experience and 165 (90.2%) were general-duty nurses. The largest number of nurses at 98 (53.65%) worked in a specialized department and 179 (97.8%) had infection management education experience. The results of the analysis of differences in the level of infection management practice according to the general characteristics of nurses, were not statistically significant (Table 1).

### 3.2. Nursing Professionalism, Time Pressure, Organizational Infection Management Culture and Level of Infection Management Practice of Nurses

The average score of nursing professionalism of the subjects was 3.29 ± 0.45 out of 5.0 points. In terms of the subordinate domains, the average scores were in the order of 3.75 ± 0.67, 3.73 ± 0.53, 3.69 ± 0.60, 2.92 ± 0.75 and 1.84 ± 0.89 for the practical nursing role, self-concept of the profession, nursing professionalism, social awareness and independence of nursing, respectively. Average scores for time pressure and organizational infection management culture were 5.35 ± 1.16 and 5.39 ± 0.79 out of 7.0 points, respectively.

The average score of the level of infection control practice was 4.56 ± 0.42 out of 5.0 points. In terms of the subordinate domains, the average scores were in the order of 4.68 ± 0.47, 4.67 ± 0.51, 4.66 ± 0.47, 4.59 ± 0.49, 4.53 ± 0.56 and 4.42 ± 0.54 for isolation, disinfection and sterilization, pneumonia control, infection control of catheter in blood vessels, urinary tract infection control and hand hygiene, respectively (Table 2).

### 3.3. Correlation between Nursing Professionalism, Time Pressure and Organizational Infection Management Culture, and the Level of Infection Management Practice of Nurses

The level of infection control practice displayed positive correlation with time pressure (r = 0.16, *p* = 0.034) and infection control organizational culture (r *=* 0.16, *p* < 0.001) (Table 3).

### 3.4. Factors That Affect the Level of Infection Management Practice of Nurses

To confirm the factors that affect the level of infection management practice of nurses, multiple regression analysis was executed with time pressure and organizational infection management culture as the independent variables. First, as the result of the verification of the presumptions of regression analysis, the Durbin–Watson statistics value was 1.72, which is close to 2, thereby fulfilling the condition of independence of residual, as there is no autocorrelation. The tolerance of independent variables was 1.00, which is higher than 0.1, and the variance inflation factor was 1.00, which is smaller than 10, thereby illustrating that there is no multicollinearity between independent variables. As the result of confirmation of the scatter plot of the residual to review the normality and homoscedasticity of residuals, there is even a dispersion around ‘0′, thereby confirming the equal variance. Moreover, normality was confirmed as points on the standardized residual normal P-P diagram are situated close to the center of the diagonal line.

As the results of the regression analysis showed, factors that affect the level of infection management practice of nurses included organizational infection management culture (β = 0.29, *p* < 0.001) and time pressure (β = 0.29, *p* = 0.024), and it was found to explain 10% of the level of infection management practice (F = 11.03, *p* < 0.001) (Table 4).

## 4. Discussion

This study is aimed at the provision of basic data for the intervention strategy to increase the level of infection management practice of nurses by assessing the factors that affect the level of infection management practice of nurses under situations in which infection management within a medical institution is emphasized due to the COVID-19 pandemic.

In this study, the average score for nursing professionalism of the subjects was 3.29 out of 5.0, while the average scores were 3.30 and 3.48 in the studies conducted on nurses by Han and Kim [15] and Kim and Park [20] using the same instrument, respectively. Meanwhile, the average score for nursing professionalism of the subjects in the study by Hwang and Lim [5] conducted on nursing students was 3.93, thereby illustrating that the nurses working in a medical institution have a lower level of nursing professionalism in comparison to the nursing students. This is deemed to be the result of the continuous learning of the nursing profession through the undergraduate educational curriculum. Therefore, it is necessary to continuously operate a program capable of improving nursing professionalism through on-the-job training, seminars and hospital ward conferences, etc., for the nurses working in clinical settings. The results of this study for the subordinate domains were in the order of the practical nursing role, self-concept of the profession, nursing professionalism, social awareness and independence of nursing, whereas the results were in the order of the independence of nursing, self-concept of the profession, nursing professionalism, practical nursing role and social awareness in the study by Kim and Park [20], thereby illustrating the difference in the results of this study. In the study by Hwang and Lim [5] on nursing students, the self-concept of the profession was the highest, while social awareness was the lowest, thereby displaying the relevance of the study subjects. Since nursing professionalism is closely related to the quality of nursing services provided to the patients [21], it is necessary to improve the independence of nursing and social awareness by providing administrative and financial support by nursing-related organizations such as the Korean Nurses Association, and government departments such as the Ministry of Health and Welfare, etc., in order to enhance the nursing professional values.

The average score for time pressure of the study subjects was 5.35 out of 7.0 while the average scores were 6.04, 5.52 and 4.96 in the studies conducted on nurses by Yun and Son [3], Teng et al. [17] and Yang, Choi, Youn & Bae [22] using the same instrument, respectively, thereby illustrating differences among studies. Time pressure is the state that the nurse experiences by being impatient to solve the problem of lack of time with the belief that there is a lack of time [23], and it is believed that the nurse experiences time pressure because he/she must perform drug administration, examination and treatment within a prescribed time, given the characteristics of nursing tasks. It is predicted that a high level of time pressure will induce negligence by the nurses on infection management tasks due to the tendency of the nurses to try to complete tasks as quickly as possible, having the perception that they do not have sufficient time. Therefore, management strategy at personal and organizational levels, including job reshuffling through job analysis and supplementation of nursing manpower, etc., is necessary to reduce the time pressure experienced by the nurses.

The average score of the infection management organizational culture of the subjects of this study was 5.39 out of 7.0, whereas the average scores were 5.56, 5.51 and 5.64 in the studies conducted on nurses using the same instrument by Lee and Park [6], Moon and Jang [11] and Kim and Song [24], respectively, thereby displaying results similar to that of this study. It is deemed that the awareness of infection management by the management within the medical institutions has increased through the fortification of the infection management domain through the revision of infection management guidelines, the Medical Service Act, and the medical institution evaluation certification due to the outbreak of emerging infectious diseases, such as MERS and COVID-19, etc. in the recent years [24], thereby elevating the level of the infection management organizational culture.

The average score of the level of infection management practice of the subjects of this study was 4.56 out of 5.0, while the average scores were 4.51 and 4.39 in the studies conducted on nurses using the same instrument by Lee and Park [6] and Hong and Park [19], respectively, thereby displaying results similar to that of this study. When examined in terms of the subordinate domains, the results were in the order of isolation, disinfection and sterilization, pneumonia management, infection management of catheter in blood vessels, urinary tract infection management and hand hygiene in this study, whereas, in the study by Lee and Park [6], isolation was the highest and infection management of catheter in blood vessels was the lowest. In the study by Hong and Park [19], pneumonia management was the highest and hand hygiene was the lowest, thereby displaying the differences among studies. It is deemed that isolation was the highest in this study due to a high level of the execution of isolation in managing the suspected or confirmed COVID-19 patients, due to the prolongation of the COVID-19 pandemic at the moment. There was no difference in the level of infection management practice in accordance with the general characteristics of the subjects in this study, as well as in studies by Lee and Park [6] and Hong and Park [19]. These general characteristics in the preceding studies [6,19] included age, educational level, position, work experience, affiliated department, infection management educational experience, etc., which are similar to those of this study. It is believed that there is a need for an additional study on the general characteristics of the subjects, and the organizational aspect is more important for the level of infection management practice than the personal characteristics of the subjects.

When the correlation between nursing professionalism, time pressure, organizational infection management culture, and the level of infection management practice of the nurses is examined, it is found that time pressure and organizational infection management culture have a positive correlation with the level of infection management practice. Studies by Lee and Park [6] and Cho and Han [7] also reported a positive correlation between the organizational infection management culture and the level of infection management practice, similar to this study. However, the study by Yun and Son [3] reported a negative correlation between time pressure and the level of infection management practices, thereby illustrating results that are opposite to this study. However, in the study by Yang et al. [22], it was found that time pressure and patient safety nursing activities have a positive correlation, thereby illustrating results similar to those of this study. This implies that an additional study is necessary to analyze the organization characteristics and situational factors, etc., of the institutions subjected to the study.

It was confirmed that factors that affect the level of infection management practice of nurses included organizational infection management culture and time pressure with the explanation power of 10%. Because the study by Cho and Han [7] found that organizational infection management culture and knowledge, and the study by Lee and Park [6] found that the level of awareness of infection management and organizational infection management culture, both affect the level of infection management practice, the organizational infection management culture was found to be a common factor that affects the level of infection management practice. The affirmative organizational infection management culture was found to be an important factor for infection management practice. Since the organizational infection management culture plays the role of motivating nurses to execute infection management activities [11], it is important to improve the organizational infection management culture first to enhance the level of infection management practices. In addition, it is necessary to establish and implement a practical infection management strategy capable of inducing the improvement of the actions of nurses through the improvement of the remuneration system, along with continuous monitoring to elevate the continuity of the level of infection management practice of nurses [6]. Therefore, in order to provide a nursing environment to the patient that is safe from infection, it is necessary to establish a well-structured infection management system within the medical institution. Moreover, it is essential to operate an infection management education program, etc., for the management and administrators tasked with practical clinical affairs to establish and build up a desirable organizational infection management culture.

The significance of this study is in having assessed the organizational infection management culture and time pressure as the results of the analysis of factors that affect the level of infection management practice of nurses under the situation of the COVID-19 pandemic. Moreover, it was possible to confirm that the affirmative organizational infection management culture plays an important role in elevating the level of infection management practice of nurses. Therefore, the management of the medical institution needs to provide administrative and financial support, and needs to improve the organizational culture in order to improve the level of infection control practice of its constituent members. It is important for the entire organization to recognize the importance of infection control and to establish an organizational culture that puts the foremost priority on infection control in order to ensure the safety of patients. However, this study has a limitation in generalizing the results since it was conducted only on nurses working at a hospital in a single region.

## 5. Conclusions

This study considered nursing professionalism, time pressure, organizational infection management culture and the level of infection management practice of nurses, and found that time pressure and organizational infection management culture affect the level of infection management practice. It also established basic data for the intervention strategy for activities aimed at enhancing the level of infection management practice of nurses at medical institutions. In order to elevate the level of infection management practice of nurses, it is important to establish an organizational culture that puts a priority on infection management within the medical institution to enable nurses to thoroughly comply with infection management guidelines at the time of executing nursing tasks. It is also necessary to provide assertive administrative and financial support by the organization in order to provide support by the management and administrators for practical clinical affairs, provision of necessary goods, continuous monitoring and feedback, etc., to enable nurses to execute their respective tasks in accordance with the infection management guidelines. In addition, it is necessary to analyze the workloads of nurses through the analysis of the duties they perform and to reduce time pressure put on nurses due to such excessive workloads. Moreover, it is important to assess the number of patients each nurse is in charge of and to rearrange the number of nurses assigned in accordance with the severity of the patients’ conditions.

The following proposals are made based on the results of this study. First, there is a need for a repeat study by expanding the hospital subjected to the study and considering the characteristics of such hospitals since the relationship between time pressure and the level of infection management practice of nurses found in this study was contradictory to the results of a preceding study. Second, it is necessary to conduct a study for the development of an administrator intervention program to elevate the level of infection management practice of nurses and confirm its effects thereof.

## Figures and Tables

**Table 1 healthcare-10-01517-t001:** Nursing Practice of Infection Control according to Participant Characteristics (*N* = 183).

Characteristics	Categories	*N*	%	Nursing Practice of Infection Control		
				M ± SD	t or F	*p*
Gender	Male	12	6.6	4.49 ± 0.46	−0.65	0.513
	Female	171	93.4	4.57 ± 0.42		
Age (year)	20~29	111	60.7	4.55 ± 0.04	1.44	0.240
	30~39	52	28.4	4.54 ± 0.45		
	≥40	20	10.9	4.71 ± 0.36		
Education	3-year college	13	7.1	4.65 ± 0.34	0.57	0.567
	Bachelor	153	83.6	4.57 ± 0.42		
	≥Master	17	9.3	4.49 ± 0.49		
Marital status	Single	131	71.6	4.56 ± 0.42	−0.30	0.764
	Married	52	28.4	4.58 ± 0.45		
Religion	Yes	50	27.3	4.53 ± 0.46	−0.59	0.559
	No	133	72.7	4.58 ± 0.41		
Work Experience (year)	<1	22	12.0	4.43 ± 0.48	1.94	0.146
	1 to <3	35	19.1	4.66 ± 0.38		
	≥3	126	68.9	4.56 ± 0.42		
Level of appointment	Staff nurse	165	90.2	4.56 ± 0.43	−0.88	0.382
	Nurse manager	18	9.8	4.65 ± 0.41		
Work unit	General ward	85	46.4	4.61 ± 0.39	1.30	0.195
	Special unit	98	53.6	4.53 ± 0.45		
Infection control Education experience	Yes	179	97.8	4.56 ± 0.43	−1.12	0.266
	No	4	2.2	4.79 ± 0.17		

**Table 2 healthcare-10-01517-t002:** Mean score for Study Variables (*N* = 183).

Variable		M ± SD	Range
Nursing professionalism		3.29 ± 0.45	1~5
	Self-concept of the profession	3.73 ± 0.53	
	Social awareness	2.92 ± 0.75	
	Professionalism of nursing	3.69 ± 0.60	
	The roles of nursing service	3.75 ± 0.67	
	Originality of nursing	1.84 ± 0.89	
Time pressure		5.35 ± 1.16	1~7
Organizational culture for infection control		5.39 ± 0.79	1~7
Nursing practice of Infection control		4.56 ± 0.42	1~5
	Hand hygiene	4.42 ± 0.54	
	Intravascular catheter infection control	4.59 ± 0.49	
	Urinary tract infection control	4.53 ± 0.56	
	Pneumonia control	4.66 ± 0.47	
	Isolation	4.68 ± 0.47	
	Disinfection and sterilization	4.67 ± 0.51	

**Table 3 healthcare-10-01517-t003:** Correlation among the Study Variables (*N* = 183).

	Nursing Professionalism	Time Pressure	Organizational Culture for Infection Control
	r (*p*)	r (*p*)	r (*p*)
Nursing practice of Infection control	0.64 (0.393)	0.16 (0.034)	0.29 (<0.001)

**Table 4 healthcare-10-01517-t004:** Factors influencing Nursing Practice of Infection Control (*N* = 183).

	B	SE	β	t	*p*
Constant	3.41	2.49		13.73	<0.001
Organizational culture for infection control	0.16	0.04	0.29	4.14	<0.001
Time pressure	0.06	0.03	0.16	2.27	0.024
R^2^ = 0.11, Adjusted R^2^ = 0.10, F = 11.03, *p* < 0.001					

SE: Standard error.

## Data Availability

The data presented in this study are available on request from the corresponding author. The data are not publicly available due to data restriction policies.

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
