# Peer review of "Factors Affecting the Infection Control Practices of Nurses at University Hospitals"

_healthcare, 2022, doi:10.3390/healthcare10081517_

Round 1
Reviewer 1 Report
First of all I would like to thank for the opportunity to review this paper. The importance of healthcare associated infection control is well known and emphasized due to the pandemic. Actually, the environmental and human factors are the main methods to their counteracting and the role of healthcare providers, especially nurses, is central.
In this context, aim of the paper under review is to assessment of factors influencing the infection control practices in a sample of nurses by using a structuralized questionnaire.
The subject under study is certainly important, especially in the historical period we are experiencing. The article presents interesting results, but it is nevertheless believed that it must be improved before publication. I would like to encourage authors to consider several issues to be improved.
Title: it can be shortened, highlight the object of the study: place, time and person.
Introduction: The authors should shorten the introduction, making clearer what is the gap in the literature that is filled with this study. The authors must better frame their study within the vast body of literature that addressed the issue of healthcare infection control and related costs (refer to articles with DOI: 10.2174/1389201020666190408095811) also at international level.
Methods: The survey was conducted using non-standard questions. The use of an unreliable instrument is a serious and irreversible limitation. The fact that a similar questions have been used in previous surveys is not sufficient. A validation process must be performed to evaluate the added questions to a standard questionnaires. What about intelligibility and validation index? Was a pilot study performed? An evaluation on the reliability has been performed, but the Authors must discuss the meaning of the different level of internal consistency between sections of the same questionnaire.
The enrolment procedure must be specified. How did the authors choose the way to select the sample? This can represent a great bias origin. How did they avoid the selection bias? How did the authors choose the way to send the questionnaire? The authors propose a minimum sample size, but what is the reference population? How large is it? Without the numerical identification of the reference population is not clear the validity of the study. A non-representative sample is by its self a non-sense-survey.
Statistical analysis: I suggest to insert a measure of the magnitude of the effect for the comparisons. Please consider to include effect sizes.
Ethical issue: it is not clear if the “Institutional Review Board of Konyang University” is a competent ethical body.
Discussion: I also suggest expanding. Emphasize the contribution of the study to the literature. The discussion must be updated in light of the economic impact of these infections (see the above mentioned reference). The Authors should add more practical recommendations for the reader, based on their findings. Also, the section of limitations and future search is also very short, the Authors could elaborate on that.
Author Response
We appreciate the opportunity to revise and resubmit this manuscript. We have read the comments of the reviewer and have revised the manuscript accordingly; we believe our manuscript has benefited immensely from these insightful suggestions for revision.

Reviewer 2 Report
Thank you for the opportunity to review the paper. The topic falls within the scope of the journal. The paper reflects the current issue of infection management, prevention and management as affected by the COVID-19 pandemic and explores factors that could influence it. I suggest a few comments for improving the overall quality.
Subjects: please provide the information about the recruitment of nurses (what method did you use?) and, if possible, include the inclusion and/or exclusion criteria
Similarly, please explain why did you choose only one hospital.
Data collection: it would be useful to describe what demographic data were used in data collection and why did you choose these
Table 1: As a reader, I am not sure what is the general ward and what the special unit - please, provide an explanation (e.g. footnote)
Table 3: it is not necessary if you mention the results in text 3.3
Overall, the study is important, and the study conclusions are supported by the results.
Author Response

(The authors gave the same response as above.)

Round 2
Reviewer 1 Report
The paper was greatly improved and it is now suitable for publication.